# Networked Cluster Formation via Trigonal Lipid Modules for Augmented Ex Vivo NK Cell Priming

**DOI:** 10.3390/ijms25031556

**Published:** 2024-01-26

**Authors:** Jaewon Park, Sungjun Kim, Ashok Kumar Jangid, Hee Won Park, Kyobum Kim

**Affiliations:** Department of Chemical & Biochemical Engineering, Dongguk University, Seoul 22012, Republic of Korea; dbffldkssk5@gmail.com (J.P.); sungjun.kim@dgu.ac.kr (S.K.); ashok4483@gmail.com (A.K.J.); hana4339@gmail.com (H.W.P.)

**Keywords:** ex vivo NK cell priming, biomaterial-mediated clustering, membrane contact, lipid-polymer conjugate, hydrophobic insertion

## Abstract

Current cytokine-based natural killer (NK) cell priming techniques have exhibited limitations such as the deactivation of biological signaling molecules and subsequent insufficient maturation of the cell population during mass cultivation processes. In this study, we developed an amphiphilic trigonal 1,2-distearoyl-sn-glycero-3-phosphorylethanolamine (DSPE) lipid-polyethylene glycol (PEG) material to assemble NK cell clusters via multiple hydrophobic lipid insertions into cellular membranes. Our lipid conjugate-mediated ex vivo NK cell priming sufficiently augmented the structural modulation of clusters, facilitated diffusional signal exchanges, and finally activated NK cell population with the clusters. Without any inhibition in diffusional signal exchanges and intrinsic proliferative efficacy of NK cells, effectively prime NK cell clusters produced increased interferon-gamma, especially in the early culture periods. In conclusion, the present study demonstrates that our novel lipid conjugates could serve as a promising alternative for future NK cell mass production.

## 1. Introduction

Natural killer (NK) cells differentiated from lymphocyte precursor cells can eliminate tumor cells or infected cells [1]. Unlike T cells, these are cytotoxic lymphocytes capable of eliminating tumors without the need for prior antigen presentation [2,3]. When inhibitory Major Histocompatibility Complex (MHC) receptors expressed on the surface of NK cells combine with MHC class 1 located on target cell membranes, the activity of NK cells is downregulated, and NK cells do not attack normal cells [4]. As MHC class 1 is reduced or absent on the surface of tumor or infected cells, the activity of NK cells increases to attack and eliminate tumor cells [5]. NK cells are able to detect antigens and secrete lytic granules, which include perforin (i.e., pore-forming cytokine allowing granzyme to enter the target cells) and granzyme (i.e., granules to activate intracellular substrates or proteins (Caspase-3, -7, and Bid protein) that can induce apoptosis of target cells) [6,7]. These NK cell functions are balanced by a variety of interactions between surface ligands/receptors. For instance, the anticancer efficacy of NK cells and subsequent tumor apoptosis could be enhanced by the interaction of tumor necrosis factor (TNF)-related apoptosis-inducing ligand receptors and Fas ligands [8].

When our body is in a homeostatic state, NK cells account for 5–15% of all circulating lymphocytes [9]. Among these, 10% are CD56^bright^/CD16^−^ NK cells, which produce interferon (IFN)-γ or TNF-α through cytokine stimulation. The other 90% are CD56^dim^/CD16^+^ NK cells, considered more mature than the former. They secrete cytotoxic granules when a combination of NK cell-activating receptor-ligand occurs [10]. Given the limited innate population of NK cells, it is important to elicit maximum activity by priming NK cells during culturing processes.

One of the conventional NK cell priming techniques is chemical priming with polyethylenimine, used for DNA complexation and transfection [11]. Another conventional method for priming NK cells involves the treatment of cytokine cocktails during the expansion process [12,13,14]. For instance, interleukin (IL)-2, IL-12, IL-15, IL-18, IL-21, and IFN-αβ are widely utilized to induce NK cell proliferation or enhance tumor cell cytotoxicity [15]. Exogenous cytokine delivery to NK cells could also provide sufficient effector functions [16]. Alternatively, people use tumor-induced cell priming, which involves the use of irradiated tumor cells as a stimulator [17,18]. However, due to the short half-life of bolus cytokines, continuous and repeated administration is required during NK cell expansion [19]. Moreover, frequent deactivation of applied cytokines in culture media often results in insufficient activation of NK cells, posing a technical obstacle for large-scale priming in the further production of autologous/allogenic cellular therapeutics [19]. Furthermore, previous studies have reported that the overexpression of immune checkpoint receptors is observed in simultaneously primed NK cells with cytokines and glucocorticoids [20]. This phenomenon could be occurred when the cytokine stimulation time is over 96 h or more [21]. Increased expression of programmed cell death (PD)-1 facilitated the binding with PD-ligand 1 on tumor surfaces and consequently reduced the therapeutic effectiveness of NK cells.

Accordingly, conventional NK cell priming techniques have exhibited a series of limitations, especially when cytokine cocktails are treated in culture media. A new priming approach is needed without any side effects, such as the deactivation of cytokines or increased immune checkpoint expression during NK cell expansion. One of the morphological characteristics of NK cells during the initial expansion is the formation of clusters. NK cells have a single-cell form when they migrate into blood vessels in the body. Whereas, during the ex vivo/in vitro expansion, the suspended NK cell population forms clusters via the development of an extracellular matrix (ECM). These architectural changes often initiate the exchange of diffusional paracrine signaling molecules and membrane-contact signal transduction within NK cells. Therefore, facilitated interaction further activates the production of cytokine activators such as IL-2. Subsequently, activated NK cells possess augmented anticancer efficacies [22].

Collectively, the enhanced guidance for the formation of NK cell clusters without cytokine or growth factor cocktails could be an ideal and beneficial alternative approach for NK cell priming. Therefore, in the present study, we suggest a biomaterial-mediated NK cell priming technique by modifying NK cell membranes with the aid of a novel trigonal-linker (TL) module (Figure 1). We attempted to form NK cell clusters by connecting NK membranes with trigonal 1,2-distearoyl-sn-glycero-3-phosphorylethanolamine (DSPE) lipid-polyethylene glycol (PEG) material (Figure 1A). The DSPE lipid anchor could be inserted into NK cell membranes via hydrophobic interaction between DSPE lipid and the lipid-bilayers of NK cell surfaces. In previous studies, comparative analyses with other lipid anchor moieties were performed, revealing that modification of the NK cell surface using DSPE imparts greater stability and prolonged maintenance [23,24]. Hence, utilizing the DSPE-lipid-based biomaterial could enhance the interconnection of the NK cell population stably and accelerate their joining to form spheroids.

Rather than stimulating NK cells with cytokines or growth factors, this biomaterial-mediated NK cell cluster formation could effectively control their interactions and regulate the signaling cascades of the NK cell population within the formed clusters. Our previous investigations [23,25,26,27,28,29,30] have proved that (1) DSPE lipid-based polymeric conjugate materials were successfully immobilized onto various cellular membranes, (2) there were no inhibitory responses in the intrinsic signaling properties of surface-modulated cells, indicating no toxicity, and (3) this biomaterial-mediated ex vivo cell surface modification could augment multiple cellular functionalities by presenting additional ligands/receptors for further cell-to-cell contacts. Furthermore, such lipid anchoring could be completed by mixing NK cell suspension and soluble TL modules at room temperature (RT) for 30 min without any additional supplements. To this end, TL-mediated NK (TLNK) cells were sufficiently primed by rapid aggregation through lipid-anchor module-dependent membrane interaction between NK cells, and augmented IFN-γ release from activated NK cell clusters was evaluated.

## 2. Results

### 2.1. Synthesis and Characterization of Trigonal-Linker Module for NK Cell Priming

To construct the TL module for NK cell clusters, DSPE-PEG-NH_2_ was conjugated with trimesic acid (TMA), resulting in (DSPE-PEG-NH)_3_-T (Figure 1A). The successful branching was verified through Fourier transform infrared (FTIR) and proton nuclear magnetic resonance (^1^H-NMR) analyses. FTIR spectra were employed to confirm the conjugation of DSPE-PEG-NH_2_ with TMA. The FTIR peak of (DSPE-PEG-NH)_3_-T observed at 3347 cm^−1^ was assigned to the N–H stretching of the newly formed amides, while the peaks at 2916, 2882, and 2850 cm^−1^ originated from C–H stretching belonging to CH_2_ groups of DSPE lipid, PEG, and CH groups of TMA. After the conjugation of DSPE-PEG-NH_2_ with TMA, the disappearance of the 1716 cm^−1^ COOH groups of TMA and the 3384 cm^−1^ terminal NH_2_ stretching of DSPE-PEG-NH_2_ indicated successful conjugation (Figure 2). Additionally, IR frequencies at 1722 cm^−1^ (ester bond >C=O stretching of DSPE-PEG), 1661 cm^−1^ (>C=O stretching of newly formed amides), 1539 cm^−1^ (C–H and C=C stretching frequencies), 1466–1343 cm^−1^ (O–H and C–H bending frequencies), 1102, 962, and 841 cm^−1^ (C–O–C stretching belonging to both DSPE-PEG and TMA) further confirmed the successful construction of the TL module.

Subsequently, the conjugation of the DSPE-PEG moieties with the TMA core was further confirmed by proton NMR analysis. TMA exhibited NMR signals as follows: phenyl ring CH protons (δ 8.66 ppm) and carboxylic group protons (δ 13.59 ppm). Similarly, DSPE-PEG-NH_2_ showed typical NMR signals of terminal methyl groups (δ 0.85 ppm), lipid chain methyl protons (δ 1.06–2.37 ppm), PEG repeating unit protons (δ 3.51 ppm), and available amide bond protons (δ 7.57–7.72 ppm). The synthesized (DSPE-PEG-NH)_3_-T revealed major NMR signals for terminal methyl groups (δ 0.85 ppm), lipid chain methyl protons (δ 0.98–2.25 ppm), PEG repeating unit (δ 3.51 ppm), amide protons of DSPE-PEG (δ 7.51–7.93 ppm), and core phenyl ring protons (δ 8.64 ppm). Furthermore, signals corresponding to the newly formed amide linker protons were observed at δ 8.43–8.57 and δ 8.75–8.97 ppm (Appendix A). These observed NMR signals confirm the successful synthesis of (DSPE-PEG-NH)_3_-T (Figure 3). Peak integration at δ 8.64 ppm (3H) relative to the terminal CH_3_ groups’ peak observed at δ 0.85 ppm (19H) and repeating units of ethylene glycol of PEG at δ 3.51 ppm (628H) further confirmed the successful conjugation of three DSPE-PEG molecules with the TMA core, resulting in the formation of the TL module (Appendix A). Additionally, by gel permeation chromatography (GPC) analysis, the average molecular weight of the (DSPE-PEG-NH)_3_-T linker was determined to be 9464 g/mol. These observations further confirmed the successful synthesis of the (DSPE-PEG-NH)_3_-T linker.

### 2.2. Viability of Surface Engineered Cells

To evaluate the activity of TLNK cells, 2 × 10^4^ NK cells suspended in media were treated with a module solution ranging from 0 to 200 µg/mL. After incubation for 6 h to form clusters, the viability of NK cells was quantified. Treatment with TL modules indicated no cytotoxicity, and subsequently, a reduced viability of NK cells was not observed, up to 200 µg/mL of TL modules (Figure 4).

### 2.3. Structural Analysis of the TLNK Cell Cluster

Typically, NK cells develop a spheroid shape starting at 24 h during initial proliferative periods. Therefore, the circularity of clustered NK spheroids at 6 h was measured to evaluate the degree of cluster formation assisted by TL modules. Each spheroid was formed with 5 × 10^3^ NK cells. Circularity is the simplest way to calculate and visualize the convex hull perimeter, which represents the roughness of the particle surface. The closer the convex hull perimeter is to 1, the smoother the surface is, and it can be interpreted that the cells form a more uniform and denser spheroid. Therefore, as a result of calculating circularity from the representative image of NK cell clusters, naïve NK spheroid without any material treatment exhibited 0.45 (Figure 5A), whereas TLNK spheroid formed by using TL modules showed 0.52 (Figure 5B). The increased circularity aided by our materials demonstrated that (1) the distance between NK cells in the spheroid is relatively close and uniform (Figure 5C), and (2) thereby the exchange of signaling molecules, as well as membrane-contact signal induction within the NK population, could be activated. Moreover, the quantification of NK spheroid volumes was performed at both 24 and 48 h (Figure 5D). Interestingly, the volumes observed in both NK spheroids and TLNK spheroids were comparable over an extended period. These results provided robust evidence that the TL module facilitates uniform cell aggregation without disrupting the intrinsic clustering function of native NK cells or influencing the physical distance between cells. Additionally, the spherical arrangement of NK cell aggregates increases the likelihood of individual constituent cells making contact with neighboring cells. Although our TL module did not have a direct impact on the macroscopic dimensions of NK cell clusters, it is conceivable that substance-mediated cluster formation could induce changes in cell-to-cell contacts, thereby potentially influencing the internal core structure of the clusters.

### 2.4. Enhanced Cytokine Release of TL-Mediated Primed Cells

NK cells activate an immune response and secrete cytokines upon the recognition of an external stimulus such as an inflammatory action [19,31,32]. As a result, the activation of NK cells can be determined by the level of cytokine (i.e., IFN-γ) secretion after treatment with inflammation-inducing substances such as lipopolysaccharides (LPS). The binding of such antigen molecules with NK cell membrane receptors (especially Toll-like receptor 4) initiates intracellular signaling pathways, upregulates the expression of IFN-γ, and thereby facilitates the exocytosis of the produced IFN-γ. When compared with naturally formed naïve NK spheroid, TLNK spheroid exhibited enhanced levels of IFN-γ secretion at both 6 and 24 h of cluster formation (Figure 6). The results demonstrated that our TL modules did not inhibit the sequential signaling mechanisms of NK cells during the priming process, including (1) membrane receptor binding with extracellularly available antigen molecules, (2) initiation of signal transduction into intracellular pathways, (3) regulation of stepwise signaling to express cytokines, and (4) secretion of produced cytokines through NK cell membranes. Moreover, a significantly increased amount of secreted IFN-γ in TLNK spheroid demonstrated that our NK cell clustering strategy using (DSPE-PEG-NH)_3_-T subsequently improved inter-membrane contacts within the NK cell population and frequent signal exchange could be obtained, as compared with natural ECM formation in naïve NK spheroid without any aid of biomaterials.

## 3. Discussion

Among a series of NK cell priming techniques, biomaterial-mediated NK cell cluster formation via accelerated membrane interaction can be a novel and effective engineering tool. This approach enables the augmented NK cell activation without the requirement of additional supplementary cytokine cocktails. Our results demonstrated that (1) TL modules could be effectively immobilized into the lipid bilayers of NK cell membranes, specifically via hydrophobic interaction [33,34], and (2) they could strengthen the membrane connections of NK cells through material-mediated physical links. In consequence, as compared with naturally-driven NK clustering, which is mediated by ECM production (without the aid of biomaterials), a significant induction of multiple membrane interactions occurred, leading to rapid and tight formation of NK cell clusters, and effective cell priming ultimately took place, resulting in enhanced IFN-γ secretion in TLNK spheroids. In addition, NK cell clusters induced by TL modules did not exhibit any cytotoxic responses or downregulation of intrinsic properties of NK cells after the administration of membrane-incorporated materials.

Such architectural characteristics of NK cell cluster formation are associated with their physiological signaling performances. Kim et al., previously compared a single NK cell culture condition, in which cells are unable to interact with adjacent neighboring cells, and a social condition that allows interactions [22]. The single-NK cell condition exhibited different downstream signaling compared to cells in the social condition. Significantly, increased interactions between NK cells improved IL-2 expression, one of the polypeptide family that mediates interactions between leukocytes. When IL-2 binds to the IL-2 receptor CD25, an intracellular signal transducer/activator of transcription (STAT) 5 is phosphorylated (i.e., pSTAT5) and further activates the expression of genes that produce perforin and granzyme B. Furthermore, pSTAT5 activation sequentially improves the expression level of CD25, activating positive feedback involved in IL-2 expression.

Additional interactions between NK cell membrane compartments also positively influence further activation of the NK cell population in clustering. For instance, several CD2 family receptors on NK cell membranes, including CD2, 2B4, CD48, CD58, and CD84, are involved in the interplay of receptors/ligands of the same one or the same family on the NK cell surface, and cells function through this interaction [35]. Particularly among these, the 2B4(CD244)/CD48 linkage plays a powerful role in the momentous effector function of NK cells, specifically IFN-γ secretion [36,37].

The secretion of IFN-γ is fatally reduced, when the absence of any one of these receptors occurs, which presumably corresponds to the decreased tumor-specific lysis function. Another study has provided evidence that the inhibition of NK cell clustering could downregulate the IFN-γ secretion ability of NK cells [38]. Furthermore, the defects in 2B4 (CD244)/CD48 led to a decrease in intracellular calcium release, which could impede perforin assembly and the exocytosis of soluble granules [35,39,40]. As such, IFN-γ is a representative indicator to evaluate the activation and priming of NK cells. In our study, as a result of TL module-mediated NK cell priming, TLNK clusters secreted more IFN-γ upon facilitated activation. The enhanced secretion level of IFN-γ is involved in activating intercellular signaling, NK cell accumulation, activation, and cytotoxicity [41]. In the process of proceeding with this signaling pathway, our TL module could avoid side effects that occur in existing conventional priming methods, such as cytokine-mediated expression of immune checkpoint receptors (e.g., PD-1), and improve the problem of high costs.

Collectively, NK cell clustering is a critical initial step of the NK cell activation and priming process through frequent and sufficient signaling with the aid of various membrane compartments. Subsequently, intracellular signal transduction and sequential gene expression are also facilitated, and the secretion of IFN-γ is enhanced in a properly primed NK cell population. To maximize the priming efficiency without the side effects of conventional priming methods, our biomaterial-mediated ex vivo NK cell priming could (1) facilitate the formation of NK clusters by connecting with surrounding NK cells through hydrophobic lipid-lipid affinity and (2) boost the rate of initial clustering by membrane-lipid interactions. Efficiently primed NK cells in TLNK clusters were examined by increased secretion of IFN-γ.

## 4. Materials and Methods

### 4.1. Chemicals

DSPE-PEG-NH_2_ (with a molecular weight (MW) of 2850 g/mol and 95% purity by GPC) was purchased from Nanosoft Polymers. Inc. (Winston Salem, NC, USA). 1-ethyl-3-(3-dimethylaminopropyl) carbodiimide (EDC, >98.0% purity) was obtained from the Tokyo Chemical Industry (Tokyo, Japan). N-hydroxy succinimide (NHS, 98% purity), di-methyl formamide (DMF, 99.8% purity) (anhydrous), 4-dimethylaminopyridine (DMAP, ≥99% purity), and TMA (95%) were purchased from Sigma-Aldrich (St. Louis, MO, USA).

### 4.2. Synthesis of (DSPE-PEG-NH)_3_-T for NK Cell Clustering

TL module ((DSPE-PEG-NH)_3_-T) was synthesized using the EDC/NHS branching reaction method [29]. Briefly, 0.1 mmol of the TMA linker (20 mg) was reacted with 0.29 mmol EDC (54.5 mg) and 0.48 mmol of NHS (54.7 mg) for carboxylic group activation (Figure 1A). The activated linker was then reacted with 0.32 mmol of DSPE-PEG-NH_2_ (849.9 mg) in anhydrous DMF, with the addition of 0.3 mmol of DMAP (37.2 mg). The reaction was stirred in the presence of N_2_ for 48 h at RT. The product was dialyzed using a dialysis tube (MWCO 6–8 kDa) against distilled water (DW) for 3 days to remove unconjugated DSPE-PEG-NH_2_, TMA, EDC, NHS, and mono/di-connected DSPE-PEG-NH with TMA. After dialysis, (DSPE-PEG-NH)_3_-T was lyophilized, yielding a final material as a white powder with a quantitative yield of 64%. The >95% purity was confirmed by the absence of intermediate signals present in NMR spectra. The compound was characterized by FTIR (PerkinElmer FTIR Spectrum Two, PerkinElmer, Shelton, CT, USA) and ^1^H-NMR (500 MHz FT-NMR spectroscopy, Bruker, Billerica, MA, US).

### 4.3. Characterizations of (DSPE-PEG-NH)_3_-T

The final synthesized TL modules for NK cell clusters were successfully characterized by FTIR and ^1^H-NMR analysis. The degree of branching was confirmed by the integrating ^1^H-NMR peaks into the core phenyl protons and terminal methyl protons of DSPE lipid. The molecular weight of the synthesized linker was confirmed by the GPC instrument (Waters, Milford, MA, USA) equipped with a Styragel GPC column using Agilent Polystyrene as the GPC standard. Tetrahydrofuran was used as the solvent, and 100 µL of the TL module solution was injected into the GPC instrument. The analysis ran at a flow rate of 1 mL/min at 35 °C for 50 min. FTIR peaks: TMA, 3087 cm^−1^ (NH stretching), 2995 and 2824 cm^−1^ (CH stretching), 1716 cm^−1^ (CO stretching), 1607 cm^−1^ (C=C stretching), 1452 cm^−1^ (C-H bending), 1407, 1272 cm^−1^ (O-H bending), and 913 cm^−1^ (C-O stretching). DSPE-PEG-NH_2_, 3384 cm^−1^ (NH stretching), 2918, 2882, 2844 cm^−1^ (CH stretching), 1733 cm^−1^ (C=O stretching), 1641 cm^−1^ (amide stretching), 1469 cm^−1^ (C-H bending), 1343 cm^−1^ (N-H bending), 1104 cm^−1^ (C-O-C stretching). (DSPE-PEG-NH)_3_-T, 3347 cm^−1^ (NH stretching of newly formed amide bond), 2916, 2882, 2850 cm^−1^ (CH stretching of both lipid and TMA), 1722 cm^−1^ (CO stretching of DSPE-PEG chain), 1661 cm^−1^ (newly amide stretching), 1539 cm^−1^ (C=C stretching), 1466 cm^−1^ (CH bending), 1343 cm^−1^ (N-H bending), 1102, 962 and 841 cm^−1^ (C-O-C stretching). NMR signals: TMA, ^1^H-NMR (500 MHz, dimethyl sulfoxide (DMSO)-*d*_6_) δ 13.59 ppm and 8.66 ppm. DSPE-PEG-NH_2_, ^1^H-NMR (500 MHz, DMSO-*d*_6_) δ 7.72 and 7.57, 3.51, 2.37, 2.26, 1.49, 1.23, 1.06, and 0.85 ppm. (DSPE-PEG-NH)_3_-T, ^1^H-NMR (500 MHz, DMSO-*d*_6_) δ 8.97, 8.87, 8.64, 8.57, 8.56, 8.54, 8.43, 7.93, 7.51, 3.51, 2.25, 1.49, 1.23, 1.17, 0.98, and 0.85 ppm.

### 4.4. Cell Culture

NK-92mi cells (gift from Doh’s lab (Seoul National University, Seoul, Republic of Korea)) were cultured with a complete growth medium consisting of Minimum Essential Medium Alpha (Gibco, Grand Island, NY, USA), 12.5% fetal bovine serum (Gibco), 12.5% horse serum (Gibco), 1% penicillin−streptomycin solution (Corning, Corning, NY, USA), 0.2 mM inositol (Sigma-Aldrich), 0.1 mM β-mercaptoethanol (Sigma-Aldrich), and 0.02 mM folic acid (Sigma-Aldrich). The cells were incubated at 37 °C (5% CO_2_, 95% humidity).

### 4.5. Ex-Vivo NK Cell Surface Engineering

NK cells were treated with (DSPE-PEG-NH)_3_-T which was dissolved in a complete growth medium at a concentration of 1 mg/mL. Specifically, 5 × 10^5^ NK cells were incubated with 100 µL of (DSPE-PEG-NH)_3_-T solution at RT for 30 min. Subsequently, the cells were washed twice with 500 µL of Dulbecco’s phosphate-buffered saline. Cells were then resuspended at a concentration of 5 × 10^3^ cells per 100 µL complete growth medium. The engineered cells were used for further experiments.

### 4.6. NK Spheroid Formation

The agarose-coated well plate was prepared by adding 50 µL of 1.5% (*w*/*v*) agarose (E&S Bioelectronics Company, Daejeon-si, Republic of Korea), which can be melted in DW using an electro microwave into a flat bottom 96-well cell culture plate (SPL Life Science, Pocheon-si, Republic of Korea) [42]; 5 × 10^3^ cells of NK or TLNK cells were then prepared according to Section 4.5 and seeded in an agarose-coated well plate. Each plate was incubated for 6, 24, and 48 h to analyze the spheroid size and quantify cytokine secretion from the NK cell cluster. The cells were incubated at 37 °C (5% CO_2_, 95% humidity).

### 4.7. Cell Cytotoxicity Analysis of (DSPE-PEG-NH)_3_-T

To analyze the cytotoxicity of (DSPE-PEG-NH)_3_-T on NK cells, (DSPE-PEG-NH)_3_-T was dissolved in complete growth medium at concentrations ranging from 0 to 200 µg/mL. 2 × 10^6^ NK cells were suspended in 200 µL of the (DSPE-PEG-NH)_3_-T solution. The suspended cells were poured into a U-bottom 96-well cell culture plate (SPL Life Science). The plate was then incubated at RT for 30 min, followed by incubation at 37 °C for 6 h. 40 µL of CellTiter-Blue^®^ (Promega, Madison, WI, USA) was added into each well for cell viability assessment. Treated cells were further incubated at 37 °C for 3 h and subsequently centrifuged at 400× *g* for 5 min to transfer the supernatant into a new flat-bottom 96-well cell culture plate. The fluorescent intensity (FI) of the supernatant was measured with a microplate spectrophotometer (Ex/Em = 560/590 nm wavelength). The percentage of cell viability was then analyzed using Formula (1):(1)$$\mathrm{Cell}\;\mathrm{viability}\left(\%\right)=\frac{\mathrm{FI}\;\mathrm{of}\;\mathrm{experimental}\;\mathrm{group}}{\mathrm{Average}\;\mathrm{FI}\;\mathrm{of}\;\mathrm{control}\;\mathrm{group}}\;\times \;100$$

### 4.8. Spheroid Size Analysis

NK cell and TLNK spheroid images were obtained using an optical microscope (Ti-E System, Nikon, Tokyo, Japan). To assess clustering efficiency (i.e., spheroid volume and circularity), characterization was conducted using ImageJ software 1.53e. The spheroid volume was calculated using Formula (2):(2)$$V=0.5\;\times \;\mathrm{Length}\;\times \;{\mathrm{Width}}^{2}$$
and circularity was calculated using Formula (3):(3)$$\mathrm{Circularity}=4\pi \;\times \;\frac{\mathrm{Area}}{{\mathrm{Perimeter}}^{2}}$$

### 4.9. Quantification of Cytokine Secretion from the NK Cell Cluster

To assess the IFN-γ released from NK cells upon inflammatory stimulation, NK spheroids were prepared following the described method in Section 4.5 and cultured for 0, 6, and 24 h. Subsequently, 1 µg/mL of LPS (from *Escherichia coli* O26:B6, Sigma-Aldrich) was administered to these cell spheroids, and the plates were incubated at 37 °C for 24 h. The IFN-γ secretion response was analyzed using an Enzyme-Linked ImmunoSorbent Assay kit (PeproTech, Cranbury, NJ, USA) following the manufacturer’s instructions.

### 4.10. Statistical Analysis

Statistical analyses were conducted for data quantification, and quantitative experiments were performed in triplicate. The results were then analyzed using a one-way analysis of variance (ANOVA) and Tukey’s multiple-comparison test on GraphPad Prism 8.0.2 (GraphPad Software Inc., La Jolla, CA, USA).

## 5. Conclusions

In this study, a biomaterial-mediated ex vivo NK cell priming technique was successfully developed, specifically via enhanced inter-membrane interactions within the NK cell population during cluster formation processes. Newly synthesized TL modules effectively initiated NK cell clustering without any inhibition in proliferation and other intrinsic cellular properties. Hydrophobic lipid insertion assembled NK cells into the clusters, which exhibited augmented activation and thereby enhanced IFN-γ production. Therefore, our material-based NK cell priming approaches and subsequently engineered TLNK cell clusters could overcome the limitations of current cytokine-based NK cell priming methods. Consequently, our result demonstrated that material-induced structural modulation in the NK cell population could be an alternative method to facilitate NK cell activation without any supplementary cytokines or growth factors.

## Figures and Tables

**Figure 1 ijms-25-01556-f001:**
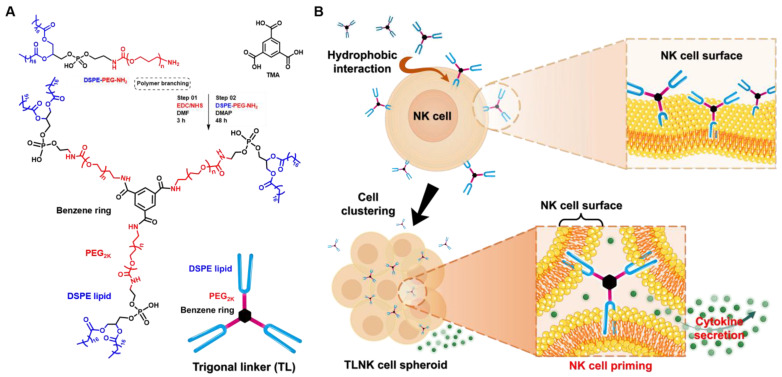
Schematic illustration of trigonal-linker (TL)-mediated NK cell clustering for ex vivo priming. (**A**) The synthesis route and complete structure of the TL module involved the conjugation of 1,2-distearoyl-sn-glycero-3-phosphoethanolamine-N-[amino(polyethylene glycol)-2000] (DSPE-PEG-NH_2_) with trimesic acid (TMA), resulting in the formation of (DSPE-PEG-NH)_3_-T. (**B**) The process of TL module-mediated NK cell clustering and priming. The upregulation of cytokine secretion, a crucial indicator of cell priming, was successfully achieved through the promotion of spheroid formation by (DSPE-PEG-NH)_3_-T.

**Figure 2 ijms-25-01556-f002:**
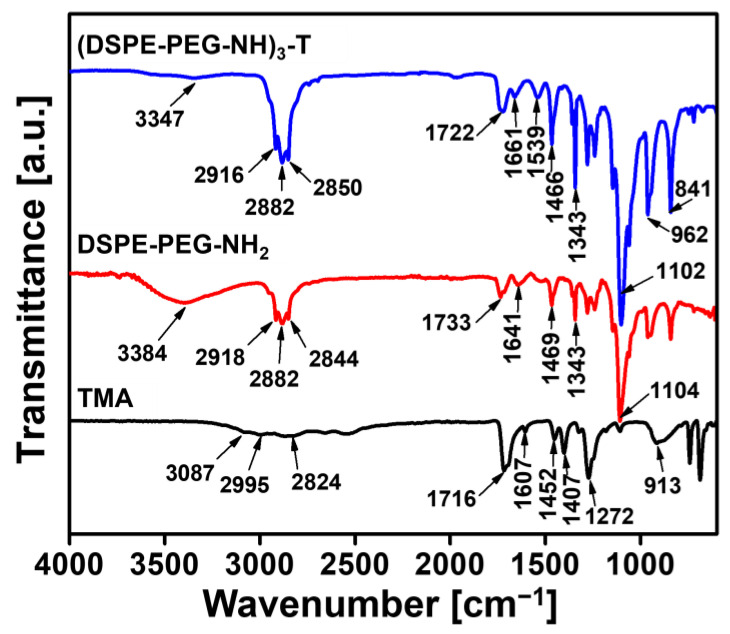
Fourier-transform infrared (FTIR) spectroscopic analysis of trimesic acid (TMA), 1,2-distearoyl-sn-glycero-3-phosphoethanolamine-N-[amino(polyethylene glycol)-2000] (DSPE-PEG-NH_2_), and synthesized trigonal-linker module, DSPE-PEG-NH branching with TMA ((DSPE-PEG-NH)_3_-T).

**Figure 3 ijms-25-01556-f003:**
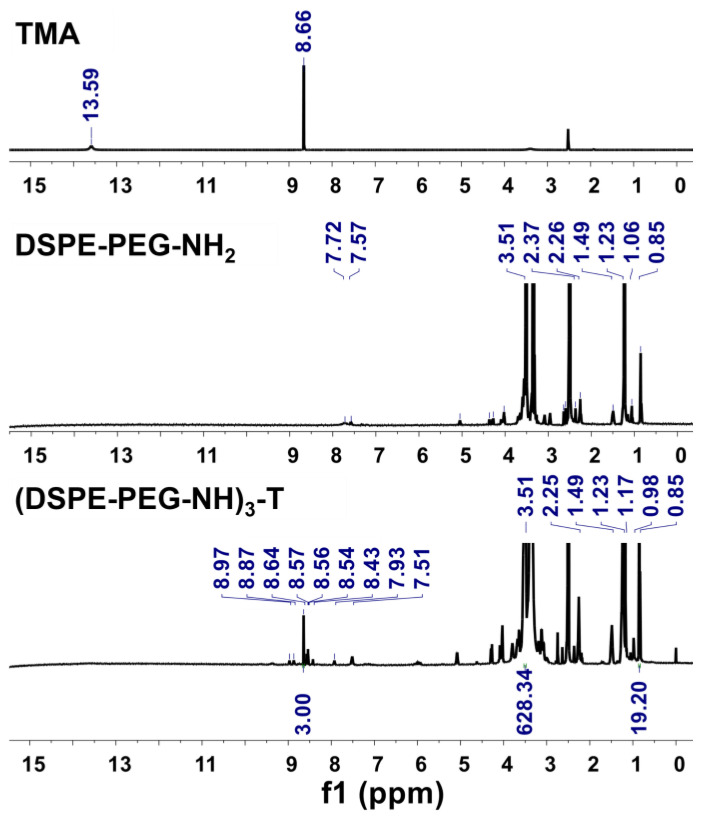
Proton nuclear magnetic resonance (^1^H-NMR) spectra of trimesic acid (TMA), 1,2-distearoyl-sn-glycero-3-phosphoethanolamine-N-[amino(polyethylene glycol)-2000] (DSPE-PEG-NH_2_), and DSPE-PEG-NH branching with TMA ((DSPE-PEG-NH)_3_-T). The NMR spectrum results were recorded in dimethyl sulfoxide (DMSO)-*d*_6_ solvent.

**Figure 4 ijms-25-01556-f004:**
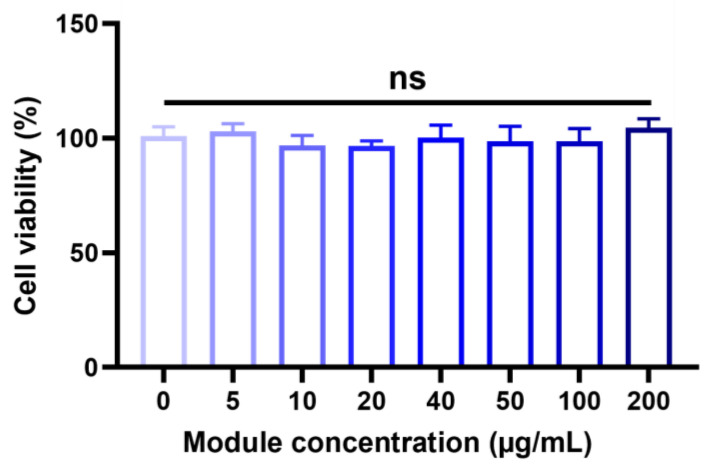
NK cell viability with the module (DSPE-PEG-NH)_3_-T for 6 h. Cell viability was evaluated in NK cells with a 0~200 µg/mL concentration of the module. In the subsequent experiment, cells were treated with (DSPE-PEG-NH)_3_-T at a 40 µg/mL concentration. The data were expressed as mean ± standard deviation (SD) based on a triplicate measurement. (ns: not significant).

**Figure 5 ijms-25-01556-f005:**
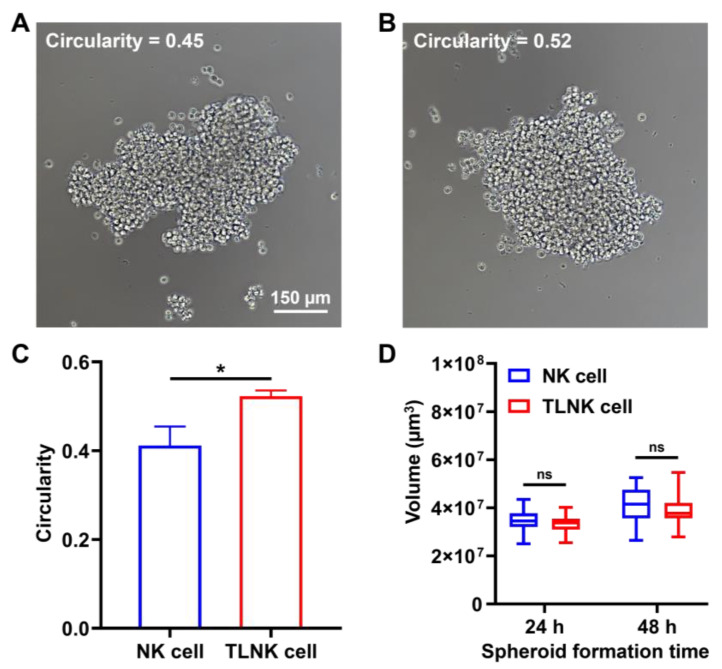
NK spheroid volume and morphology analysis. NK spheroid formation image at 6 h in (**A**) the NK cell group and (**B**) the TLNK cell group. (**C**) Cell circularity with (DSPE-PEG-NH)_3_-T for 6 h. (**D**) Volume quantification with (DSPE-PEG-NH)_3_-T for 24 h and 48 h. A single spheroid was made up of 5 × 10^3^ NK cells. The area, perimeter, length, and width of spheroids were measured by ImageJ software 1.53e. Also, the spheroid circularity and volume were calculated with Formulas (2) and (3). The data were expressed as mean ± standard deviation (SD) based on a triplicate measurement. (* *p* < 0.05, ns: not significant).

**Figure 6 ijms-25-01556-f006:**
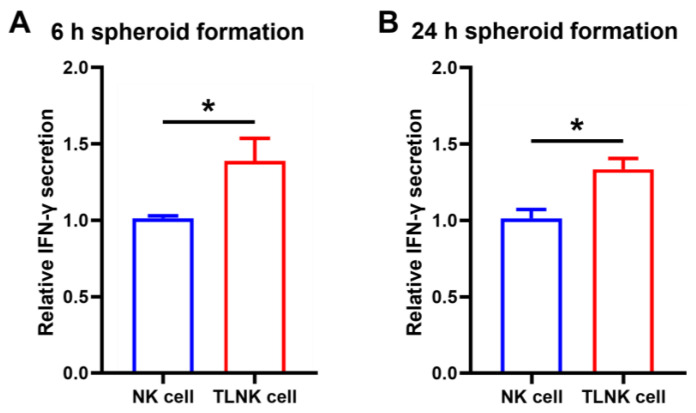
IFN-γ secretion in TLNK cells after (**A**) 6 h and (**B**) 24 h of spheroid formation. All groups were treated with 1 µg/mL lipopolysaccharide (LPS) for 24 h for cytokine inducement. IFN-γ secretion was detected by ELISA and the data are expressed as mean ± standard deviation (SD) based on a triplicate measurement. (* *p* < 0.05).

## Data Availability

The data supporting the findings of this study are available within the article and its Appendix A.

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
