# Peer review of "Networked Cluster Formation via Trigonal Lipid Modules for Augmented Ex Vivo NK Cell Priming"

_ijms, 2024, doi:10.3390/ijms25031556_

Round 1

Reviewer 1 Report

Comments and Suggestions for Authors

After these following minor revision, the manuscript can be accepted.

·       In the introduction part, there is lack in the logical flow between the paragraphs, which can be improved.

·       Numerous articles have been published prior on this tip; therefore, the significance of this work should be reinforced and justified.

·       Figure 1 can be improved.

Comments on the Quality of English Language

·        There are numerous grammatical errors in the manuscript, which require further improvement.

Author Response

Please see the rebuttal and revised manuscript. 

Reviewer 2 Report

Comments and Suggestions for Authors

Developing amphiphilic trigonal DSPE-PEG linker for assembling NK cell clusters is no doubt a good approach. Authors have demonstrated that synthesized trigonal linker is not cytotoxic and showed enhanced cytokine release. However, there are some concerns regarding the synthesis of these linkers.

1) What was the molecular weight of DSPE-PEG-NH2 authors have used to synthesize the linker?

2) How did authors check the purity of synthesized linker?

3) What was the molecular weight of synthesized linker? Since it is a lipidic polymer, authors can use MALDI or GPC to detect the molecular weight of synthesized linker.

4) FTIR and 1H NMR results have been repeated. Line #113 to 139 explains the FTIR and 1H NMR observations for linker. Authors have repeated it in section 4, subsection 4.3.

5) Do check spelling mistakes in manuscript. E.g. in line #107, “trional” should be “trigonal”.

Comments on the Quality of English Language

Do check spelling mistakes in manuscript. E.g. in line #107, “trional” should be “trigonal”.

Author Response

(The authors gave the same response as above.)

Reviewer 3 Report

Comments and Suggestions for Authors

In the current study, the authors utilized trigonal lipid modules to enhance the assembly of natural killer cells, presenting various characteristics to validate the formation of TL modules and TL-modified NL. They assert that material-induced structural modulation within the NK cell population could serve as an alternative means for facilitating NK cell activation without additional cytokines or growth factors. However, the showcased characteristic measurements are insufficient to substantiate their conclusions, and the essential links between these characteristics and physiological signaling performances are absent. Consequently, I cannot endorse the publication of the work in its present state, further, the writing style requires additional refinement. More comments are given below:

1.     One big issue of the study is that multiple characteristics measurements have been listed, such as the preparation of TLNK, circularity, IFN-γ secretion, etc,. But these results do not have further interpretation and lack of connections between these characteristics and physiological signaling performances, as later introduced in Section 3.

2.     Continue with the above question, the volumes of TL-modified and normal NK cell groups are the same, so what is the possible mechanism of the enhanced activation levels? Besides the apparent changes in IFN-γ secretion, what else could we expect from TL-modified NK?

3.     The current research includes only 29 references, with at least 6 of them originating from the authors themselves, which falls short of an adequate scholarly citation count. Notably, the paragraph spanning lines 65 to 68 lacks any external references, exemplifying the insufficient incorporation of relevant literature.

4.     The paragraph between lines 128 to 139 is not consistently written. For example, the authors started with “to investigate the degree of (DSPE-PEG-NH)3-T”, and ended with “confirmed the successful formation of the TL module”. Where is the qualification of the degree of (DSPE-PEG-NH)3-T (the degree of (DSPE-PEG-NH)3-T also needs explanation)? 

5.     The lack of symmetry in the error bars, where they do not consistently extend equally above and below the bars, introduces confusion.

6.     The x-label for Figure 4 has format issue, please double check.

7.     The data quality in Figure 5 is not satisfying, especially in Figure 5d.

Comments on the Quality of English Language

Moderate editing of English language required

Author Response

(The authors gave the same response as above.)

Reviewer 4 Report

Comments and Suggestions for Authors

Manuscript: ijms-2815818

The manuscript of Jaewon Park, Sungjun Kim, Ashok Kumar Jangid, Hee Won Park and Kyobum Kim “Networked cluster formation via trigonal lipid modules for augmented ex vivo NK cell priming” presents the studies regarding the development of amphiphilic trigonal 1,2-distearoyl-sn-glycero-3-phosphorylethanolamine lipid-polyethylene glycol for assembling natural killer cell clusters via multiple hydrophobic lipid insertion into cellular membranes resulted in promising alternative for future natural killer cell mass production.

The readability and the structure of the manuscript are good, and the research design is appropriate. However, there are some issues that need to be added and addressed before publication. I am not a strong expert in the biological part of the work, my comments mainly will be addressed in practical terms.

Major issues:

1.      Please improve the quality of the structures of compounds in Figure 1.

2.       Please justify the choice of 1,2-distearoyl-sn-glycero-3-phosphorylethanolamine for NK cell membrane modification.

3.      I can't agree with the author's explanation of 1H NMR spectra based on data from Figure 3. At first, spectra were recorded in different solvents: DSPE-PEG-NH2 in deuterochloroform, while TMA and (DSPE-PEG-NH)3-T in deuterated dimethyl sulfoxide. It is not correct to compare chemical shifts of the signals obtained in different solvents.  Secondly, the spectrum of (DSPE-PEG-NH)3-T does not meet the requirements, and the signal maxima are not visible. I wish the authors to explain signals at the interval 5.00-8.00 ppm in the 1H NMR spectrum of (DSPE-PEG-NH)3-T. Authors wrote,Additionally, the peaks observed at 8.43 to 8.52 ppm, 8.87 and 8.97 ppm signals belong to multiple amide protons available in the synthesized (DSPE-PEG-NH)3-T…” Please, also provide integration of these mentioned signals of amide protons. Please add in the supplementary file full 1H NMR high resolution spectra for each mentioned compound. In the manuscript, authors could include spectra only of the fragments confirming structural changes of the compounds.

4.      Please add information about the final purity of the obtained (DSPE-PEG-NH)3-T.

5.      Please justify the purity of used reagents in section 4.1.

6.      In section 4.2. authors wrote,The compound was characterized by FTIR (PerkinElmer FTIR Spectrum Two, PerkinElmer, USA), and 1H NMR (500 MHz FT-NMR spectrometer, Bruker, Germany)”. Please add the corresponding FTIR and 1H NMR data.

Minor issues:

1.      Line 81. Please use the lowercase “d” instead of “D” in the name of 1,2-Distearoyl-sn-glycero-3-phosphorylethanolamine.

2.      Please check and correct the abbreviation μg/mL in Figure 4.

Author Response

(The authors gave the same response as above.)

Round 2

Reviewer 1 Report

Comments and Suggestions for Authors

Manuscript is acceptable. Regards,

Author Response

Please find the attached rebuttal and revised manuscript. 

Reviewer 2 Report

Comments and Suggestions for Authors

Authors have adequately addressed the comments in the revised version of the manuscript. I will recommend to accept the manuscript. 

Author Response

(The authors gave the same response as above.)

Reviewer 3 Report

Comments and Suggestions for Authors

All my initial concerns have been addressed properly, thus, I agree the publication of the study in IJMS

Author Response

(The authors gave the same response as above.)

Reviewer 4 Report

Comments and Suggestions for Authors

Please write  the NMR spectra data correctly, mentioning the multiplicity of signals.

Author Response

(The authors gave the same response as above.)
